# Neurology Meets Theology: Charles Sherrington's Gifford Lectures Then and Now

## Michael A. Flannery

UAB Libraries, University of Alabama at Birmingham, Birmingham, AL 35294, USA; flannery@uab.edu

**Abstract:** Charles Scott Sherrington (1857–1952) is widely acclaimed as the most important neurophysiologist in history. He became a legend in his own time, coined the term "synapse", and in 1932 received the Nobel Prize in medicine for his discoveries on the function of neurons. By the time he presented the Gifford Lectures 1937–38, he represented the best that science had to offer on behalf of the relationship of the mind to the natural world. The lectures, including one never publicly presented, were published as *Man on His Nature* (1941). Here neurology meets theology at the busy and often treacherous intersection of science and religion. Examining Sherrington's views in some detail, the standard rendering of Sherrington as a theist cannot be sustained by their contents; he ends up as at least a humanist and perhaps an atheist. Views by neurologists and philosophers of mind some seventy to eighty years later are compared and contrasted with Sherrington's. Although expectations of a materialist/reductionist answer to the mind/body problem have not been realized, neuroscientist Raymond Tallis appears as a parallel figure to Sherrington: both are clearly naturalistic humanists. A theistic response is presented addressing the mind/body problem from a hylomorphic process theology perspective, along with some comments regarding natural theology in general. In the end, this essay has two complementary aims: (1) to relocate Sherrington's neurotheology—if it can be called that—in a more appropriate historiographical category; and (2) to offer a viable answer to the mind/body problem.

**Keywords:** John C. Eccles; Albert Einstein; Ragnar Granit; Wilder Penfield; hylomporphism; natural theology; neurology; neurotheology; process theology; mind/body problem

## 1. Introduction

Sir Charles Scott Sherrington (knighted 1922) was born in Islington, London on 27 November 1857, and lived a long and fruitful life that by its end on 4 March 1952, earned him the appelation, "the William Harvey of the nervous system" (Jellinek 1994). Far from a purely scientific background, his early education was steeped in the classics. But his professional interests began when he entered Cambridge University, studied under famed physiologist, Sir Michael Foster, and graduated with first class honors in 1882. From 1882 to 1884 he performed cortical localization studies while a medical student with John Newport Langley, but soon, under the influence of Walter Gaskell, he shifted his attentions to the spinal cord (Molnár and Brown 2010, pp. 431–32). This gave him the necessary background to eventually develop a more complete picture of the brain and nervous system, including the key concept of synapse (the transmission of nerve impulses between neurons), a term he coined with Foster in 1897.

In 1904 Yale University invited him to give the Silliman Lectures, which "did for the nervous system what William Harvey's *De Motu Cordis* did for the circulatory system" (Eccles and Gibson 1979, p. 19). The Silliman Lectures were published one year later as *The Integrative Action of the Nervous System* in which the principles of adaptation under natural selection previously applied with such success to morphology could now be applied to neurophysiology; the principle that "pure reflexes are admirably adapted to certain ends" was established (Granit 1966, p. 54). This publication opened Sherrington's work to a

much larger audience, placing him on the right-hand of such a luminary as Harvey himself; it is not by coincidence that Sherrington gave the introductory speech at the Tercentary Celebration of Harvey's *De Motu Cordis* at the Royal College of Physicians in London, 1928 (Eccles and Gibson 1979, pp. 217–20). It might be said that *The Integrative Action of the Nervous System* demystified the nervous system; after this publication, if there was a "ghost in the machine" it would not be found there.

Space precludes detailing all of Sherrington's accomplishments. It will have to suffice that from 1913 until his retirement in 1935 he held the Waynflete Professor of Physiology Chair at Magdelen College, Oxford, but not before receiving the 1932 Nobel Prize in Physiology for his pathbreaking discoveries on the function of neurons. It should be noted that Sherrington's love for the humanities (born of his early training in the classics) never wavered, and it was this common interest that probably nurtured his friendship with medicine's paterfamilias, William Osler, who was influential in Sherrington's appointment to the chair of physiology at Oxford. Thus, Sherrington was well prepared philosophically and professionally to give the Gifford Lectures in Natural Theology (1937–38), the subject of this essay.

Before launching into their content, a few words concerning the structure of these lectures are in order. The first set of lectures were delivered at Edinburgh University in May 1937 with the second set given in June the following year. The first group comprised chapters one through six, the second group were continued as chapters seven through eleven. The last lecture was never publicly presented, having been cancelled due to the impending coronation, and will be presented as a postscript in part 4. These were all compiled and published in 1941 as *Man on His Nature*. Taken altogether, given the importance of Sherrington at the time of their presentation, it is easy to see why the Editorial Advisory Board for the Classics of Medicine chose this as a select volume in their series, writing, "*Man on His Nature*, also the result of lectures, is considered Sherrington's pivotal work as 'the supreme philosopher of the nervous system'" (Bogdonoff et al. 2000). Science and theology would meet at its most fascinating intersection—mind and body—and the remainder of this essay will examine whether the Board's assessment is deserving.

## 2. Historical Context

Finish-Swedish neuroscientist, Ragnar Granit, himself a Nobel Laureate (1967) who had worked with Sherrington at his lab in 1928 and returned there as a Fellow of the Rockefeller Foundation in 1932–33, gives an interesting picture of the state of neuroscience during Sherrington's youth. At that time it was not understood that nerve fibers were always connected or associated with nerve cells, and French neurologist Alfred Vulpian "had hazy notions about 'granular matter' of the spinal cord", while Moritz Schiff's keen observation that grey matter is the site of reflex activity remained to be rigorously proved (Granit 1966, p. 27). Even Michael Foster's (1876) *Textbook of Physiology*, which Sherrington read at Cambridge, presented the nervous system as a "protoplasmic network" connected by impulses of varying resistance. "Foster thus", as Granit puts it, "for very good reasons, avoided speaking in precise terms. None were available". Sherrington changed all that. In other words, "one might say that the reflexology Sherrington encountered in his early studies was an experimental science based on gross anatomy.... The nerve cell with its interconnections became his analytical unit. This was Sherrington's first great notion, one of his main contributions to neurology" (Granit 1966, pp. 28–29).

Neurology was thereby permanently transformed, and moreover, a paradigmatic transformation of Kuhnian proportions was wrought by Sherrington. He was able to associate the cortex of the brain with certain movements in the precentral gyrus (those involving voluntary movements) and the postcentral gyrus (those involving involuntary movements), and his research at least partially confirmed the dynamic polarization of the nerve cells postulated by Ramón y Cajals in papers published in 1891 and 1897. In many ways he represents for brain and neuroscience what Albert Einstein does for physics. The Sherrington model of neurophysiology was one of complex and layered synaptic

interconnections, a system without mysterious entelechies or vague "fluid" forces, just as with Einstein's universe of space-time relativity permanently transformed our view of the universe.

Since our current appreciation for the basic operations of the neurological system owes its foundation to Sherrington, his opportunity to hold forth on its implications for natural theology might well answer the question, what happened when neurolgy met theology? In fact, Sherrington became a "figurehead" for science itself, the embodiment of those urbane and high culture aspirations characteristic of the undaunting search for truth—honest, forthright, and directed towards the betterment of humankind—once the undisputed territory of the theologian and now since Darwin and the secular evangelism of his X Club apostles given over to a new priestly caste, the scientist. Lord Adam Gifford sought to reconcile the two with the lectures that bear his name.

It has been pointed out that Sherrington was presenting in an extremmly fertile intellectual environment, a period when "writing on life and mind was rich in creative, and varied, thought on these subjects. Freud was writing *An Outline of Psycho-Analysis*; Koffka had just published his work on Gestalt psychology; Schrödinger was writing on life in terms of energy and entropy; and C. S. Lewis was writing of entities which were not material" (Harrison 1968, p. 320).[1] Einstein, of course, might well have been added here. Although he continued to resist the implications of quantum theory, by the time of Sherrington's Gifford Lectures his iconic status in science was well established; he had received his Nobel Prize more than a decade before and had relocated to Princeton to escape from the rise of Nazism in his homeland. If mind and body (biology) form the north/south intersection of science and religion, it might be said that the cosmos (physics) forms its east/west crossroad. Before investigating Sherrington's north/south path, it might be worthwhile to pause and ask, what was Einstein's east/west direction? It appears to turn in a pantheistic direction, or so K. W. Harrison believes. He quotes Einstein at length:

> I maintain that cosmic religousness is the strongest and most noble driving force of scientific research. Only the man who can conceive the gigantic effort and above all the elevation, without which original scientific thought cannot succeed, can measure the strength of the feeling from which alone such work can grow. What a deep belief in the intelligence of Creation and what longing for understanding, even if only of a meagre reflection in the revealed intelligence of this world, must have flourished in Kepler and Newton, enabling them as lonely men to unravel over years of work the mechanism of celestial mechanics. . . Only the man who devoted his life to such goals has a living conception of what inspired these men, and gave them strength to remain steadfast in their aims in spite of countless failures. It is cosmic religiousness that bestows such strength. A contemporary has said, not unrightly, that the serious research scholar in our generally materialistic age is the only deeply religious human being. (Harrison 1968, p. 326)

Einstein was one of those larger-than-life "saints" of science who played a significant part in the cultural life of the modern world (if only by his rank and public standing); a leader of his field who was at the same time "a firm believer in transcendence" (Dyson 2006, p. 16). But, in the end, he merely turned science into a kind of religion, which had an unfortunate impact on narrowing his vision (Dyson 2016). This kind of reductionism led to his fruitless quest to find a unified theory based upon an elusive set of equations (Dyson 2006, pp. 10–11). The extent to which Einstein's views reflect those of Sherrington's will be worth investigating.

## 3. The Lectures Then

It might be helpful to start not with the content but with the responses to *Man on His Universe*. Many of our initial impressions of these lectures and the book that followed have come from those who knew him; without exception they view him as a dualistic theist. Wilder Penfield (1957), an exceptional student of Sherrington's who ushered in modern

neurosurgery, for example, declared, "Nature spoke to him with many tongues and with accents many. He understood their accents and could integrate them in his thinking until he seemed at last to comprehend the meaning of life, the design of the Creator" (p. 410). Similarly, John C. Eccles (1989), another protégé of Sherrington's who received the Nobel Prize in 1963 for his work on cell membranes in the peripheral and central nervous systems, paid almost worshipful homage to his old mentor, saying:

> I have great empathy with my old master, Sherrington, in his deeply moving message [quoting from the final chapter, "Conflict With Nature"]. I believe that biological evolution is not simply chance and necessity. That could never have produced us with our values. I can sense with him that evolution may be the instrument of a Purpose, lifting it beyond chance and necessity at least in the transcendence that brought forth human creatures gifted with self-consciousness. (p. 116)

Ragnar Granit (1966), already mentioned, saw Sherrington differently but still as more or less as a dualistic theist of sorts: "His is a Natural Religion and if a name for describing his standpoint be wanted, it is 'Evolutionary Pantheism'" (p. 174). The sum of these assessments is that Sherrington's lectures amounted to a kind of neurotheology.

Of course each of these men came with certain *a priori* dispositions. Penfield, a Presbyterian, performed a series of experiments in which he could, by using high-tech equipment, artificially raise a subject's arm. Then, by asking the subject to raise his own arm, Penfield compared the results. Most interestingly, the subject could tell the difference between the involuntarily induced arm movement and the voluntary one. What explained the subject's voluntary movement? Whence this intentionality? The involuntary movement was obviously prompted mechanically by Penfield, but the voluntary movement was mind-motivated. These studies, he concluded, suggested that consciousness was independent of physical conditions that were necessary but not sufficient to explain mental processes (Penfield 1975). Eccles had a much more complete metaphysic that echoed the teleological evolution proposed by natural selection's cofounder, Alfred Russel Wallace (1910), in his *World of Life: A Manifestation of Creative Power, Directive Mind and Ultimate Purpose* nearly eighty years earlier (Flannery 2018, pp. 136–40). Granit's position was less explicit but inclined towards a nonreductionist view of the cosmos. He agreed with the neo-Darwinian account of evolution, but thought it was an incomplete explanation. The scientist, he believed, needed to treat questions in deep time such as the origin of the universe and the origin of life with humility. Echoing Einstein, he had a reverent and "religious attitude toward the unknown" (Margenau and Varghese 1992, pp. 177–78).

The question is, can these conclusions be legitimately drawn from Sherrington's own statements as published in his *Man on His Nature*? Do they put words into Sherrington that aren't there? Do these statements really represent nothing more than confirmation bias on their part? Answering these questions requires examining Sherrington directly.

Sherrington frames his lectures from beginning to end almost as a dialogue with the mid-sixteenth century physician, Jean Fernel. Of particular interest to Sherrington is Fernel's *On the Hidden Causes of* Things (1542). Sherrington indicates great respect for the man he considers the first serious physiologist. For Fernel, nature is under God's direction, "the study of Nature is a study of final causes; divine intentions" (Sherrington [1941] 1999, p. 38). But Fernel, to Sherrington's admiration, was a rationalist viewing nature as all-encompassing, an inquisitive and perceptive scientist who tested the empirical claims of astrology and found it wanting and resisted the superstitions of his age by maintaining a commitment to a version of substance dualism before Descartes. Having seen what today would be called psychiatric disorders, Fernel wanted to know if they were preternatural. Sherrington ([1941] 1999) notes that, "He [Fernel] is inadequately aware of the importance of the 'self' in the natural history of the mind. Nor would he get much help from Aristotle in this matter. He could have got more from St. Augustine of Hippo. Of course it was still a century to Descartes and the famous 'cogito ergo sum'. That, looked at broadly, seems to mark, besides a parting of the ways in philosophy, a starting-point in mental

medicine" (p. 44). Fernel believed that the mind was part of physiology not because he was a materialist but quite the opposite, because he thought that *all* bodily operations were immaterial, animated by a vital force. Thus Fernel presents a conflicted view: on the one hand man is fully a part of nature, on the other hand, acts of the body were beyond the laws of nature being part of the immaterial soul. Throughout his lectures, Sherrington juxtaposes Fernel's valiant attempts to understand the human body and its relation to nature with the knowledge of mid-twentieth century science, which may have been an intentional rhetorical device (Granit 1966, p. 170); the essence of which is that so much of what Fernel thought of as immaterial we now know today as due to empirically testable, material phenomena.

This raises the question of why Sherrington introduced Fernel in the first place. One obvious answer is that he was already familiar him. The Gifford lecturer was a thorough-going bibliophile of Oslerian proportions and so it is not unlikely that he chanced upon the Renaissance physician during book-hunting excursions or perhaps even earlier as a student learning of Fernel's discovery of the fluid-filled canal in the spinal cord (Ambrose 2022, pp. 77–78). Sherrington had told his student Eccles as he prepared the lectures that he used Fernel as a means to "relieve the spiritual bleakness of so much of his story by introducing little vignettes from the Age of Faith as represented by Fernel", especially making the entire presentation more palatable to his clergy-filled audience (Eccles and Gibson 1979, p. 112). Zeman (2007) considers Sherrington's use of Fernel "an ingenious tactic. Fernel's views, sympathetically presented, provide an historical perspective on more modern theories of life and mind... to remind us, *sotto voce*, how radically the views of humane and intelligent people on these topics have changed over the ages" (p. 1984). In that sense, Fernel might well represent an exemplary foil falling into Sherrington hands much as William Paley was used by Darwin to contrast his views with the famed creationist (Shapiro 2014). Whatever the reason, Sherrington ([1941] 1999) certainly agrees with his Renaissance counterpart that humans are part of nature (pp. 361, 384, 389) and he has sympathy for Fernel's dualism (pp. 319, 324, 325, 335, 344). Mind is *not* material for Sherrington. It is not just the brain inside your head, it is something more, but what exactly that something is remains unclear. He seems to suggest that mind is an emergent property accrued through Darwinian survival mechanisms affording early hominids certain selective advantages and in that sense a unity of purely natural forces (Sherrington [1941] 1999, pp. 143, 201).

Therefore, those who have claimed that Sherrington was always hesitant and uneasy with dualism are correct (Zeman 2007; Ambrose 2022, p. 78). His brand of dualism did not lead toward natural theology at all. As Granit (1966) puts it, "Sherrington's dualism is a purely pragmatic acceptance of energy and mind as phenomena of two categories" (p. 172). But even this can be taken too far. Sherrington is not, despite his admiration for Fernel, a typical substance dualist. If anything he is vaguely reminiscent of the dualistic psychic and physical parallelism of Spinoza (Boodin 1934, pp. 252–53).[2] "The course of scientific discovery", he insists in his final lecture for the 1937 half of the series, "has since then conspired with this view of Descartes to cut the individual into two disparate halves, mind and body. That severance is pronounced a paradox by Nature and Evolution. Nature and Evolution deal with the individual, body and mind together as a unity" (Sherrington [1941] 1999, p. 185). He calls the attribution of an immortal soul to human beings a "trespass" that "ruptured" the unity of the whole person by "a fateful amplification of the psyche" from the finite "persona" that is reserved only for "revealed religion", a consideration he refuses to enter into (Sherrington [1941] 1999, pp. 337, 355). With mind itself reserved to a general category of nature, then, it seems reasonable to conclude with Eccles and Gibson (1979) that we are left with little more than "an unsatisfying appeal to panpsychism" (p. 154). This is a dualist panpsychism without God, or at the very least without a First Cause, that resolves only into Nature (usually capitalized to denote ontological significance) as a cosmic reality (Sherrington [1941] 1999, p. 348). He ends his last presented lecture, "Two Ways of One Mind", by asking a provocative question: "Between these two, naked mind and the perceived world, is there

then nothing in common? . . . They have this in common", he answers, "they are both concepts; they both of them are parts of one mind. They are thus therefore distinguished, but are not sundered [presumably as revealed religion has done]. Nature in evolving us makes them two parts of the knowledge of one mind our own. We are the tie between them. Perhaps we exist for that" (Sherrington [1941] 1999, p. 357).

## 4. "Conflict with Nature", a Postscript

One last unpresented lecture remains, the final chapter to *Man on His Nature*; it gives not a message of unity but of "Conflict With Nature". Instead of offering evidences for God in nature as expected of any good Gifford lecturer, he sketches instead a brief treatise on human suffering through the "life history of the malaria parasite, teleological proof of the existence of the Devil" (Costello 1941, p. 362). Life—"a crowning curse"—is not sacred for Sherrington; there is too much suffering in the world. Nature is indifferent to humankind. He even defends materialism, calling it "an inspiration for dealing with Nature" (Sherrington [1941] 1999, p. 365). Yet there is a clear value in this, transcendent in its own way. With magic exorcised from Nature, we can now appreciate its tragedy and its comedy for what it is, "a harmony all its own.. . . When Cleanthes speaks of compensation for pain—there is here compensation. It is the old primeval gift of knowledge, which we, wiser now, know was not primevel but is of yesterday—therefore with promise of further [growth and expansion]" (Sherrington [1941] 1999, pp. 400–1).

Sherridan ends by suggesting that Natural Religion cannot give us the things that the great revealed religions can give. The mystical religions of the East and the Abrahamic faiths of revelation provide movement and transfiguration of vast communities. It infuses its adherents with necessary emotion, "zest for life" that nothing else can give. But Nature *can* bestow values. Beauty is one. Wonder is another. More importantly, Natural Religion, like the great religions of the world, holds truth as an enduring value. Science, by shedding the heavy anchor of anthropomorphism, is the constant colleague of Natural Religion. In the end, beauty, wonder, and truth yields the numinous. If we no longer have a "higher mind" or a "higher personality" to lean on and obtain direction and counsel from, then left only with the capacities of our evolutionary emergent mind, we are called to "loftier responsibility" amidst this life of "tragedy and pathos" (Sherrington [1941] 1999, p. 404).

## 5. Assessment

Can a label be applied to any of this? Although Sherrington was not a materialist, he obviously had great respect for the problem-solving abilities of methodological naturalism in science. While a dualist of sorts, he was certainly not a Cartesian substance dualist. Although not a religious man, he was not irreligious. Whatever else might be said of Sherrington's lectures, it should be obvious that the strong theism (or at least teleological implications) suggested by Eccles and Penfield is simply not there. Many simply took Sherrington at face value. The well known and popular anthropologist Ashley Montagu (1941), for example, considered it "a wonderfully good book", going so far as to call it "the one indispensable book which every educated person of our time should make it his business to read, and to read slowly and carefully, for few books have been published in our time as important as this. *Man on His Nature* is destined to have a profound effect upon the philosophy of the present and the future" (pp. 544–45). Similarly, but perhaps a bit less exuberantly, H. B. Adelmann (1942) called it a "deeply learned but thoroughly fascinating work by the dean of British physiologists" (p. 227).

Others, however, were less enthralled. University of Liverpool zoologist William Ellis (1941), author of *The Idea of the Soul in Western Philosophy and Science* (1940), thought Sherrington's lengthy excursions into factual scientific description left a mass of "raw undigested material" (p. 198). He further questions a main argument of Sherrington's "that if we had a complete understanding of the laws of chemistry and physics as exemplified outside living organisms we would have at our disposal a complete apparatus for the prediction of events inside the organism" as if the laws of the external world would unveil

the secrets of organic life internal to it (Ellis 1941, p. 202). "To the best of my belief", he continues, "no one has ever succeeded in adducing the behaviour of proteins from that of carbon, hydrogen, nitrogen and the other elements into which the protein molecule can be resolved" (Ellis 1941, p. 203). In fact, Ellis finds Sherrington's constant association—one might say conflation—of the inorganic with the organic a persistent feature throughout *all* of *Man on His Nature*. With telling rhetorical accuracy Ellis asks, can we even reduce the problem of suffering to "a detailed description (such as may be found in any text-book of biology) of the life history of the malaria parasite?" (Ellis 1941, p. 207). In the final analysis, Ellis finds Sherrington's knowledge expansive but philosophically superficial and metaphysically vacuous. Harry T. Costello (1941), Brownell Professor of Philosophy at Trinity College, complained, "seldom have the Gifford Lectures on natural theology contained outwardly less natural theology than this series" (p. 359). Like Ellis, Costello argued that Sherrington mistook the *necessary* conditions of chemistry and physics as *sufficient* to explain a functioning organism, a judgment inadequate to explain the intricacies of biology, much less the complexities of mind (Costello 1941, p. 361). Recent immigrant to the United States, Viennese psychiatrist Rudolf Allers (1941) (mentor to Viktor Frankl), thought Sherrington's presentations couldn't have been what Lord Gifford had in mind in establishing the lectures because what was offered was an unsoliticited "Theology of Nature, as if the task were to replace the Divine by Nature" (p. 507). Allers found Sherrington far from the urbane, knowledgeable polymath many have claimed him to be. Instead he indicted the founder of modern neurological physiology as ignorant of the fact that the sixteenth century read Aristotle through Aquinas's eyes. Pointing out many of Sherrington's historical misinterpretations, Allers (1941) concluded, "They reveal a certain farness, a lack of comprehension, in spite of the best will, a fact not uncommon with scientists when they approach matters of history or speculation. The scientific mentality creates, it seems, a peculiar disposition which bars the way to a real understanding of history and everything but pure science" (p. 508). Harsh words, but K. W. Harrison concluded much the same. For whatever reason, Sherrington seems reluctant to speculate with any confidence, and yet that is precisely the entire reason for the Gifford Lectures in the first place. Ultimately, his views on natural religion and evolution are questionable (still laboring under simplistic notions of "survival of the fittest" and "Nature red in tooth and claw"), leaving his strength "in analysis rather than synthesis" (Harrison 1968, p. 319).

Given the validity of all these weaknesses, the question of labeling Sherrington's ideas remains. It is an important one because the criticisms summarized above then become critiques of the identifier thus applied. Sherrington is surely a naturalist. He believes along with most naturalists that nature is all there is or at least all that we can know. But he does not dismiss the transformative power of religious belief; even if the traditional religions fail to have "real" content, they call us to higher purpose. One is reminded of Robert Browning, whose religious views are as hard to pin down as Sherrington's, in his poem "Andrea del Sarto" (1855): "Ah, but a man's reach should exceed his grasp,/Or what's a heaven for". Like Sherrington, humanity rather than god (lower case intended) is the primary focus. God exists for *our* purposes. Thus, we can say two things about Sherrington's philosophy: first, he is a naturalist; second, he is a humanist. Whether he is religious or not depends upon how one chooses to define that term. And it seems that Sherrington may have been presenting during a significant shift in the concept of natural theology itself, a time when Christianity was losing its strength and standing allowing for a "dazzling array of free-thinkers" to mount the Gifford lectern, such that "the natural in 'natural theology' is potentially so broad that any configuration of atoms in the universe, or any human idea, could be the subject of critical reflection, and the question of what makes the reflection theological, and therefore in keeping with the intentions of Gifford, is a persistent one" (Birch 2022a, pp. 60, 63).

Whatever the implications for Gifford Lectures as a whole, there remains a compelling and revealing parallel between Sherrington and Einstein. As already pointed out, Ragnar Granit believed Sherrington to be an evolutionary pantheist. Similarly, Einstein has been

legitimately considered a cosmic pantheist. But this raises questions. Schopenhauer once said that pantheism is a mere "ephemism for atheism" because it becomes a "self-defeating proposition" in which God's otherness is lost to the universe. Pope Pius XI declared that Einstein's theory of relativity was "authentic athesim even if camouflaged as cosmic pantheism" (qtd. in Rubenstein 2022, p. 23). It has been asked with some force, "how can his [Einstein's] cosmic divinity be unchanging and absolute if the space-time it amounts to is dynamic and relative?" (Rubenstein 2022, p. 28). Einstein's cosmic God of unity and order is nothing like his cosmology of flux, change, and relativity (Rubenstein 2022, p. 30). Even Mary-Jane Rubenstein (2022), herself sympathetic to pantheism, admits with some equivocation that Einstein's pantheism may come down to a kind of atheism (p. 33). The same may be said of Sherrington, though not surprisingly it is more biologically focused on evolutionary emergence. The great bond that Sherrington has with Einstein is their almost worshipful attitute towards science. We have seen it clearly expressed in Einstein, but it is in Sherrington too. In humanity's "striving after Truth" in order to better our own condition, our "adventure into Natural Science is one aspect of that same endeavour. There would seem therefore to be between Natural Theology and Natural Science in some degree an aspiration in common, and in some measure a potential ground on which both stand" (Sherrington [1941] 1999, p. 4). It may be said, then, that Sherrington's religious humanism/naturalism may be considered an effective atheism because when the surface is penetrated only a severely ontologically diminished god is there, a reference to scientific endeavor in pursuit of human betterment is all that remains. God and science can certainly coexist, but not when the ontological categories are shifted from God and science to Science and god, whether that god resides in the cosmos (as for Einstein) or that god resides in the immaterial mind of *H. sapiens* (as for Sherrington).

But more than eighty years have passed since these Gifford Lectures were presented. How might Sherrington be responded to now? The best way to address this is to meet Sherrington on his own ground—science. What does neurology tell us about scientific naturalism today?

## 6. The Lectures Now

The great divide between Sherrington's neurology and our own was proclaimed when President George W. Bush along with Congress announced on 17 July 1990, the Decade of the Brain. Along with this much acclaimed proclamation came unprecedented funds for research into the three pounds of grey matter encased in our skulls in the hopes of revealing the secrets of the human mind. Advances in digital techniques such as functional magnetic imaging (fMRI), Positron Emission Tomography (PET), and Computed Tomography (CT) were thought to now show the brain as it functioned in real time thus unlocking the secrets of its mysterious operations. It was even presumed that such devices could help locate the so-called "God spot" in our brains, the very the seat of our religiosity.

But intense investigation failed to turn up anything of the kind. Neuroscientist Mario Beauregard, who had done postdoctoral work at the Montreal Neurological Institute at McGill University established in 1934 by Wilder Penfield, and Beauregard and O'Leary (2007) reported, "Our findings demonstrate that there is no single 'God spot' in the brain located in the temporal lobes. Rather our objective and subjective data suggest that RSMEs [religious, spiritual, or mystical experiences] are complex and multi-dimensional and mediated by a number of brain regions normally implicated in perception, cognition, emotion, body representation, and self-consciousness" (p. 272). Furthermore, experiments on Carmelite nuns indicated that deep mystical states could be reached by their intense contemplation in recalling and "reliving" previous mystical experiences (Beauregard and O'Leary 2007, p. 274). This was unexpected. In addition, unlike in Sherrington's day when the central dogma of early neuroscience was that neurons were fixed and unchanging at a certain age, it is now recognized that an important principle is that the brain can reorganize itself in "neuro plasticity"; nothing else in the human body shows this kind of malleability. As Beauregard and O'Leary (2007) point out, "the few traditional simplicities in neuro-

science are vanishing. The brain turns out to be more like an ocean than a clockwork" (p. 105). They conclude that the fundamental mistake is to assume along with a host of materialist advocates such as Daniel Dennett, Richard Dawkins, Susan Blackmore, Patrica Churchland, and others that mind *is* brain. There is an "immense epistemological gap between the psychological realm (*psyche*) and the physical realm (*physis*)" such that "*psyche* cannot be reduced to *physis*" because they "represent complementary aspects of the same underlying principle; neither can be entirely discounted in favor of the other" (Beauregard and O'Leary 2007, p. 292).

By the end of the 1990s the much touted "Decade of the Brain" revealed many interesting aspects of neurological activity, but utterly failed to explain how the brain turned the world "out there" into many component parts and then, even more mysteriously, how it integrated them back together again into thoughts, perceptions, attitudes and actions, and especialy into the unique attributes of qualia, the phenomenal properties of personal experience. Throughout the 1990s, "at every turn the neuroscientists found themselves completely frustrated in their attempts to get at *how the brain actually works*" (Le Fanu 2009, p. 17). Indeed for all the scientific advance since Sherrington's lectures—which was real and significant—understanding of the mind/body problem has witnessed little progress. Even the editor of *Nature*, John Maddox, admitted, "We seem as far from understanding [the brain] as we were a century ago. Nobody understands how decisions are made or how imagination is set free" (qtd. in Le Fanu 2009, p. 19). The human genome project similarly raised more questions than it answered. If, for example, Bonobos share 98 percent our genome, how do we explain the profound differences between the lower primate's mental capacity and our own? (Le Fanu 2009, p. 190). Sherrington ([1941] 1999), like Darwin, saw only differences in degree rather than kind between animals and humans, calling "man as one of the animals" one of "the most fruitful" steps biology had taken (p. 140). For all of Sherrington's rejection of anthropomorphism, he failed to see how it could also improperly explain away human exceptionalism. A more recent review of the situation suggests the scientific failures of over anthropomoprizing animals. Bolhuis and Wynne (2009) write:

> Over the past two decades, researchers have reported that chimpanzees can empathize with other members of their species, and that they reconcile and even console each other after conflicts. Monkeys and apes have been credited with a sense of fairness and aversion to inequity and, in the case of apes, an awareness of the mental states of others—in other words, a theory of mind.
>
> A closer look at many of these studies reveals, however, that appropriate control conditions have often been lacking, and simpler explanations overlooked in a flurry of anthropomorphic overinterpretation. For instance, capuchin monkeys were thought to have a sense of fairness because they reject a slice of cucumber if they see another monkey in an adjacent cage, performing the same task, rewarded with a more-sought-after grape. Researchers interpreted a monkey's refusal to eat the cucumber as evidence of "inequity aversion" prompted by seeing another monkey being more generously rewarded. Yet, closer analysis has revealed that a monkey will still refuse cucumber when a grape is placed in a nearby empty cage. This suggests that the monkeys simply reject lesser rewards when better ones are available. Such findings have cast doubt on the straightforward application of Darwinism to cognition. Some have even called Darwin's idea of continuity of mind a mistake. (p. 832)

The problem of interpreting continuity of mind between species and of understanding mind itself is not nearly as straighforward as sometimes supposed. But Sherrington's humanism would not allow him to yield completely to the kind of reductionist errors described above. He admitted to a "residue" in nature—the conscious "I"—the source of all our meaningful values encompassed in mind and energy. This indeed seems a unique attribute of the human "animal" and we needn't ascribe human-like qualities to our nearest biological cousins to appreciate that; this is a genuinely Sherringtonian sentiment.

Nevertheless, whatever else may be said between Sherrington's neuroscience of mind and today's, James Le Fanu's (2009) thesis that "two of the most ambitious scientific projects ever conceived [the Human Genome Project and the Decade of the Brain] have revealed, quite unexpectedly—and without anyone really noticing—that we are after all a mystery to ourselves" (p. xv).

These so-called "mysteries" are not so disturbing for physician and clinical neuroscientist Raymond Tallis. An unapologetic atheist humanist, Tallis doesn't concern himself with human-to-animal continuities. For him, "our bigger brains did not merely give us more of the same but helped to facilitate something qualitatively different; why we are not just very bright chimps" (Tallis 2016, pp. 225–26). Tallis rejects this kind of reductionism, calling it "Darwinitis". Furthermore, he takes aim at what he calls "Neuromania", the idea that we are nothing but our brains. These dogmatic faiths of our age, he argues are, "two key intellectual pathologies that have a large and growing presence in the realm of ideas—in academe; in the wider republic of letters; and in the popular imagination" (Tallis 2016, p. xii). Although religion has drawn at least as much blood as it has inspiration, *that* is not Tallas's primary concern. It is scientism and the reductionism that flows from it that is the pox upon our age; these are keeping us from realizing our potential. He even admits, "Things must be pretty dire when even an atheist like me wants to rescue, if not God, at least the idea of Him (or Her or It). But it's true. Neuromaniac and Darwinitic approaches to religion do such inadequate justice to the most profound, and possibly the most terrible, idea mankind has ever entertained, that I feel almost protective towards the Old One" (Tallis 2016, p. 327). Though with more philosophical accumen and wit, he sounds like Sherrington. Tallis singles out Patricia Churchland, academic extraordinaire and "Queen of Neuromania", for setting the stage for the Decade of the Brain's bankruptcy with *Neurophilosophy* (1986), proclaiming with all her hearfelt evangelical zeal that the sum of our moral standards and values can be found in neurobiology (Tallis 2016, p. 317). Against Churchland, Sherrington's cautious approach to human values seems a more prudent wisdom.

Tallis rather than arguing against human exceptionalism embraces it. Like Sherrington, the human mind is an emergent property of evolutionary processes that is much more complex than mere survival of the fittest. Although Darwinian evolution contains some central truths for Tallis, the creativity found in the human brain cannot be explained in those simplistic terms. Cumulative selection; niche construction; epigenetics; developmental biology; phenotypic plasticity; geographical distribution and climate along with many other factors have combined to create the remarkable capacities of the human intellect. While much has been lost in a fruitless search for the ghost in the machine, we now have with equal unfrutifulness focused solely on the machine. Tallis's thoroughly nondogmatic and nonreductionist approach suggests that we not succumb to the idea that leaving religion behind obligates us to the kind of naturalistic excesses found in Neuromania and Darwinitis. Tallis (2016) regards the human spirit as more than the material world, and if we are to understand what it is to be human we must embark upon that "spiritual adventure" (p. 361). We do not need God or angels to show the way, but neither will we find it in killing off our own spirit by reducing it to the lockstep of dogmatic Darwinian processes or the mechanistic neurological systems within our bodies. This spiritual adventure must include not just science but all the humanities—art, music, drama, poetry, literature, philosophy, history, etc.

If there is a current Sherringtonian today it is Raymond Tallis. Tallis (2016) has called the Gifford lecturer of more than eighty years ago, "the greatest neurophysiologist of all time" (p. 20). He praises Sherrington for his cautions against "the doctrine of localization" (perhaps the key failing of the 1990s) in favor of a more integrative approach, and in his analysis of how the evolutionary development of bipedalism in early hominids freed the arms and hands from the requirements of locomotion, he quotes Sherrington's "beautifully expressed" passage from his landmark publication, *The Integrative Action of the Nervous System*, that our forelimbs were transformed from "a simple locomotor prop to a delicate

explorer of space" (Tallis 2016, p. 216). Sherrington's lectures—rather awkwardly situated within the context of natural theology—make them more hesitant and at times more confused than Tallis's less encumbered humanism. Whether Tallis can be fully accepted as an atheist rather than as a religious humanist depends upon one's definition of *religious*. We have seen, of course, Sherrington's own atheistic tendencies. Although Sherrington and Tallis are both naturalists, they are not reductionist naturalists who outright reject religion. Sherrington would likely nod in tacit agreement with Tallis when he states, "To naturalize religion is to naturalize even those parts of humanity that are most remote from the natural world. It is the supreme expression of a devastating reductionism that disgusts even an atheist like me. In defending the humanities, the arts, the law, ethics, economics, policies and even religious belief against neuro-evolutionary reductionism, atheist humanists and theists have a common cause and, in reductive naturalism, a common adversary: scientism" (Tallis 2016, p. 336).

## 7. A Theistic Response

Most theists would see two problems with the humanism of Sherrington and Tallis. For the theist this ignores the more important question of divine relationship with humanity. Secondly, leaving the unique capacities evidenced in the mind of *H. sapiens*—human exceptionalism (self-awareness, abstract thought, artistic endeavor, even worship itself)—to the blind operations of evolutionary forces (be they Darwinian, neo-Darwinian, or something else) seem not to be the inference to the best explanation because we have no example of how such intentional complexity could ever be generated by blind natural phenomena. If, as Tallis suggests, they arose through the emergence of bipedalism thus (recalling Sherrington) freeing the forelimbs to reach for the starry beyond, then this begs the question, whence bipedalism in the first place? We see bipedalism in other animals (everything from bears, kangaroos, and the great apes to monitor and basilisk lizards) but no concomitant emergence of anything even approaching the human mind. Evolutionary biology has proposed various answers, but they are all speculative and metaphysically unsatisfying. A lurking realist bias suggests—without confirmatory evidence—that because a naturalistic answer is available, it must be somehow be better than the more idealist answer. The realist perspective that all phenomena emerge from preexisting certainties still begs the question of their existence in the first place.

In addition to these difficulties with humanism, the real question being raised by Sherrington is the mind/body problem; here the theist has much more to offer than the problematic substance dualism of Descartes or parallelism of Spinoza. Seventy-three years after Sherrington, Anglican philosopher Roger Scruton gave his own Gifford Lectures and presented a far richer theological picture amidst a far more entrenched atheistic culture. While the current wave of atheism springs from many motives, one of the more important is, in Scruton's (2012) words, "the desire to escape from the eye of judgement", a move accomplished by eliminating the personality of people and of God—escaping "the eye of judgment by wiping away the face" (p. 2). Stated another way, as we have already seen, science has offered explanations of the moral life by dismissing or explaining away human exceptionalism. "Such would-be explanations", insists Scruton (2012), "assimilate human to animal conduct only by giving the most superficial descriptions of both. In particular they leave out of consideration the radically different *intentionality* of the human response" (p. 28). By reducing human beings to the world of objects—of neural pathways, of genetic determinants, of evolutiuonary survival and adaptation, of things—you are no longer describing people but "a world from which human action, intention, responsibility, freedom and emotion have been wiped away: it will be a world without a face" (Scruton 2012, p. 49). In this sense, religion is at heart communitarian, based upon interaction and communities of belief. It is not by mere whim that the Eucharist is Christianity's deepest personal and communal expression. Scruton ends his lectures by pointing out that the removal of God from our cultural landscape has come all too easily by replacing the numinous and transcendent experience of the divine with a consumerism of appetites,

defacing of *eros*, and the elimination of the old conception of life within the community and togetherness with a rapacious individualism of self-gratification. No wonder, then, "that moments of sacred awe should be rare among us. And surely it is this, rather than the arguments of the atheists, that has led to the decline of religion. Our world contained many openings onto the transcendental; but they have been blocked by waste" (Scruton 2012, pp. 177–78). If the Gifford Lectures seem increasingly given by those least interested in it, we might look to Scruton's conclusion, not to scientistic hand-waving, for the answer.

But it is not in Scruton's Gifford Lectures that the mind/body problem is extensively addressed, but rather in the Stanton Lectures, delivered to the divinity faculty of the University of Cambridge and published as *The Soul of the World* (2014). One of Scruton's key points is that there is no way to describe the subject/I experience simply by neural pathways. Cognitive science offers little explanation for how truth-claims and reference claims arise and how these *a priori* reflections take place almost instantaneously, without anything like systematic research and/or scientific method. What we are always left with are promissory notes of purported future discovery; not science but wishful thinking. Scruton (2014) suggests a cognitive dualism (not to be confused or conflated with the ontological dualism of Descartes or the murky parallelism of Spinoza) whereby we can understand the world "in two incommensurable ways, the way of science and the way of interpersonal understanding" (p. 33). In cognitive dualism "we can grasp the idea that there can be *one* reality understood in more than one way" (Scruton 2014, p. 66). Thus, as Scruton suggests, every human being is indeed two things—an animal and a person—the one is explanatory, biological (of science), the other is understanding, "aboutness"—the "I" of self-awareness—metaphysical (of philosophy).

In effect, there are two epistemological ways—very different from Sherrington's "two ways of one mind"—there is "the way of explanation, which searches for natural kinds, causal connections, and universal covering laws, and the way of understanding, which is a 'calling to account,' a demand for reasons and meanings" (Scruton 2014, p. 184). It would be absurd to suggest they are not related. Even Tallis and Sherrington would agree with Scruton that personhood is an emergent feature of being human, just as music is an emergent feature of auditory wave lengths or art is an emergent feature of color pigments, each necessary to the other but not reducible to either. To confuse or conflate the two, however, is something on the order of a category mistake. Our personal *encounter* with God is at heart an I-thou relationship of *understanding* rather than an I-it relationship of *explanation*, perhaps making the excursion into natural theology something of a fool's errand (Moser 2017, pp. 324–28). But an interesting reconciliation of the two ways is available. Here Scruton makes an important addition by arguing that Aristotle's theory of hylomorphism sheds light on this issue by presenting "the view that the essential nature of an individual thing is given by the concept underwhich its parts are gathered together in unity" (Scruton 2014, pp. 68–69). This fits together with his cognitive dualism by appreciating what science has to offer while at the same time acknowledging songs as well as sounds, art as well as color, John and Jane Doe as well as their mass of neural synapse.

Scruton's invocation of hylomorphisim is particularly useful, an ancient but important theory that requires some further unpacking. This Aristotelian concept comes from combining two Greek words, *hyle* (matter) and *morphe* (form). Sherrington opened his lectures by pointing out that everything is composed of properties we ascribe to chemistry and physics, but this misses the essential hylomorphic insight: the difference is not in the matter *qua* matter. "The hylomorphic has no need to say there's some additional thing present in living matter but absent in nonliving things. Both cases present us with matter characterized by form, and the difference is in the form and not the matter" (Davison 2018). Form, insists Anglican Priest Andrew Davison, is neither independent of matter nor can it be defined only in terms of matter. Life is a formal characteristic not a material one. Hylomorphism was advanced in a Christian context by Thomas Aquinas in his *Summa Theologiae* (composed 1265–1274). Today hylomorphism has resurfaced particularly among Thomistic theologians such as Edward Feser, Eleonore Stump, and Andrew Davison (cited

above). Hylomorphism's contribution is that it seems to relieve the tensions arising from the mind/body problem. The problem with hylomorphism as typically construed is that its forebearers (Aristotle and Aquinas) posited a very autocratic diety imposing form upon matter by divine fiat. This translated almost organically into a Christian context of sheer power. Gifford lecturer from the previous decade (1927–28), Alfred North Whitehead (1978), stated, "The Church gave unto God the attributes that belonged exclusively to Caesar" (p. 342). As philosopher John Elof Boodin (1934) more historically if less poetically observes:

> If the conception of the Hebrew prophets had been dominated by the feeling for righteousness and the Greek conception by the feeling for beauty, it was inevitable that the Roman conception should be dominated by the feeling for power. It is true that St. Augustine, the authoritative interpreter of Christianity, tells us that "good created the world"—a reminiscence of Platonism. But the attributes of God which give the fundamental tone to his theology and the theology of the Church are omnipotence and omniscience. There must be no limitation to God's power and knowledge. (p. 437)

Aquinas carried this idea forward, and it is this kind of imposition of order with which Einstein unwittingly envisioned his more abstract god, a universe directed by an Unmoved Mover that ironically had little resemblance to his own cosmos of constant change and space-time relativity. What is needed is a deity, placed in a hylomorphic context, that is itself more dynamic and interactive with creation.

Process theology provides this, and as we shall see, at the same time solves the mind/body problem. To begin with, Charles Hartshorne (second only to Alfred North Whitehead in developing process theology), supported a cosmic order of absolute freedom between creature and Creator by pointing out Einstein's difficulty. "The only God I think I can conceive", he said, "is the one Einstein rejected, a 'dice-throwing God'" (Hartshorne [1937] 1975, pp. ix–x). The process perspective emphasizes being as becoming in mutual interaction between God and creation; there is an ongoing nature to time of which God is a part. Everything is interrelated and interdependent, and all of existence consists (in Whiteheadian terms) of events or occasions of experience.[3] More significantly, everything has absolute freedom or agency, a freedom that limits the omniscience of God because deity works preveniently in collaboration with the free choices of the created world, especially those of his creatures. God is, therefore, characterized by the process theist as omnipresent but not all-knowing since there is no way for even the most divine to know what free choices—for good or ill—will be made.

Now it might seem odd to fit Aristotle's Unmoved Mover and Thomas's omniscient and absolute God into such a process framework. Richard Rorty (an early proponent of process thought) and his teacher Hartshorne, however, argue that Aristotle was wrong to suggest that God exists totally apart from the world of moving things. As Hartshornian philosopher Daniel Dombrowski (2007) puts it, "To be God is to be the living form of the world's matter, 'the *energia* and *entelechia* of its *dynameis*.' God would be nothing if God were not the essential factor *of the world*, specifically the harmony of the natural ends of particular things in the world . . . So also, on a consistent version of cosmic hylomorphism . . . God would be immanent in the world as its intelligible order even *if* God would also *in a sense* transcend the world as its ideal end. God as an Unmoved Mover (or the gods as unmoved movers) ruins this internal-external balance" (pp. 90–91). In other words, there is no pantheism, only *panentheism* which is quite different. These are apparently nuances that neither Einstein nor Sherrington could see or appreciate. Instead, Einstein flirts with a pantheism that ends up as atheism and Sherrington struggles with a hesitant dualism of energy and mind reduced to an evolutionary unity that is itself another form of atheism. The real philosophical issue here is how does God relate to creation—the one and the many. Indeed, how might Scruton's communitarian nature of religion itself be fully realized?

Process theologian and Jesuit priest Joseph A. Bracken offers an answer. Citing Whitehead's (1933) *Adventures of Ideas*, Bracken (1991) points out that "The real actual things that endure are all societies. They are not actual occasions.. . . Thus a society, as a

complete existence and as retaining the same metaphical status, enjoys a history expressing its changing reactions to changing circumstances. But an actual occasion has no history. It never changes. It only becomes and perishes" (p. 45). One might say that actual occasions are "governed by a personally ordered subsociety of living occasions" (see note 3) which do not have mind but agency that goes up the scale of complexity. These collective agencies should be regarded as societies rather than substances. Bracken's solution to the problem of the one and the many is this: "the material universe is a structured society that is itself included in the trinitarian society of the three divine persons. Thus the ultimate onological reality is a society, not an individual entity (or even a compound individual entity)" (Bracken 1991, p. 49). Even at lower levels of agency—genes and cells—the structured body as a whole functions intergratively with the soul. "The body does not exist for the sake of the soul nor the soul for the sake of the body. Both exist for the sake of the composite that exercises activity as a unitary whole. In Whiteheadian terms, the actual occasions constituting the body are imminant within the successive occasions of the regnant subsociety and vice versa" (Bracken 1991, p. 48). As such, efforts to find localizations of mind—whatever aspects they may be—are misguided. In fact, "whether the dominant occasion is to be found in this part of the brain or some other part at any given moment is irrelevent since, wherever it is located, it contributes its individual agency to the collective agency of the organism as a whole" (Bracken 1991, p. 48). We needn't struggle over matter and mind because increased complexity arises from actual occasions, preserved into enduring societies building greater levels of emergent agency and complexity with no need for divine intervention—just as ontogeny recapitulates phylogeny—with the Trinity providing what Whitehead calls "divine initial aims" within nature itself. Like matter (*hyle*) and form (*morphe*), spirit is everywhere as an intrinsic part of nature but its self-organization and complexity varies (Bracken 2008, pp. 117–18). It is rather reminiscent of the neural plasticity mentioned earlier, an ocean not a machine.

The idea of Bracken's trinitarian cosmology and its mind/body implications has been unwittingly but cogently expressed even by otherwise secular philosophers by pointing out that "hylomorphists deny that brains generate beliefs, desires, and other mental states. Brains do not generate mental states. . . any more than wood and metal generate pianos. Rather, brains are subsystems that enable humans to interact with each other" and express themselves as belief, desire, pleasure, pain, etc. as "phenomena [that] are not causal byproducts of lower-level neural processes; they are rather patterns of social and environmental interaction that comprise lower-level phenomena—ways in which lower-level neural processes can be structured or organized. Experiences [Whiteheadian occasions] cannot be inner events reflecting the outer world, they are in fact patterns of interactions involving people, properties, and events in the world" (Jaworski 2011, pp. 308–9, 317). These are best understood as *societies* constituting *social* interactions.

Process-based hylomorphic theism has no need to posit dualism of any kind, and it avoids question begging emergence—though accepting emergence in principle—by the initial aims of the Trinity working with a perfectly free creation guided by a divine lure. This new form of "emergence" drawn from physics, chemistry, and biology "all mutually appeal for a new and more comprehensive worldview in which the relation between spirit and matter will be dramatically reconceived and redefined" (Bracken 2008, p. 126). It is surely a cosmic dance of success and failure, but it retains the freedom of genuine creativity in the process. From this standpoint, even evolution—however construed—is always much more than mere survival advantage and blind conditions of nature, it must itself be imbued with a creativity that even the cofounder of natural selection Alfred Russel Wallace recognized.

## 8. Conclusions

All of this defies a definite conclusion. Even the most religiously sympathetic investigators admit that nothing can prove the existence of God, only that the "inadequate theories of RSMEs concocted by materialists" can be ruled out (Beauregard and O'Leary

2007, p. 277). But as we have seen, Sherrington or his later manifestation in Tallis, could not be considered materialists. They *are* properly regarded as humanists. Adding the adjectives religious or atheistic to the term is more difficult. It is hard to determine on the sliding scale of subjective belief where any given humanist may lie. The "facts" we amass in support of our repective worldviews (theistic, agnostic, or atheistic) defy the certainties of science. Even Paul admitted: "For now we see in a mirror dimly" (1 Cor. 13:12 [ESV]).

Sherrington and Tallis show the difficulties involved. Both have real respect for the religious perspective; Sherrington, though less explicit than Tallis, is probably at bottom an atheist, but this conceals more than it reveals. A lurking epistemic problem remains for humanism as expressed by Sherrington and Tallis or anyone else. As Hartshorne ([1937] 1975) observes, "Humanism condemns us to a lack of integration within knowledge itself . . . To say nature is godless is to say that it is not basically intelligible. The only thing that fully explains itself to a purposive rational mind is a purposive rational mind; everything else suggests the need for explanation" (p. 23). Scruton agrees. With Sherrington we get no explanation; with Tallis we get speculation. Both are inadequate.

In the end, Sherrington's Gifford Lectures reveal more about the direction of the lectures themselves. As already mentioned, we can dispense with the fiction that Sherrington found God in nature, and if so we might wonder along with Jonathan Birch (2022b), "whether there is an unworlding faction on the Glasgow Gifford committee working for the abolition of natural theology through a process of appointing lecturers who have nothing to say on the discipline" (pp. 72–73). But even negation is more than nothing, and it might even be said that Sherrington pointed in some way toward the future. Curiously, just as Sherrington was presenting his humanistic version in Edinburgh, Karl Barth was giving a very different theistic answer in his Gifford Lectures in Aberdeen, *both* curiously enough pointing away from natural theology.

Loyola University philosopher Paul K. Moser has more recently suggested theological and exegetical reasons for dispensing with natural theology. Paul appeals "to a direct experience of God's intervening Spirit, in the experience of divine love" that is rooted in personality—the I-thou relationship—varying and tailored to each relationship; "Paul does not allow for the dilution of this unique evidence from God by any argument from natural theology" (Moser 2017, pp. 327–28). But Moser's rejection of natural theology rests upon its narrow and superficial interpretation as an apologetic proof rather than as an aspect of a divine lure towards deeper relationship. Whitehead (1978) states, "we must investigate dispassionately what the metaphysical principles, here developed [in the philosophy of organism—i.e., process thought], require on these points, as to the nature of God. There is nothing here in the nature of proof. There is merely the confrontation of the theoretic system with a certain rendering of the facts . . . [God's] unity of conceptual operations is a free creative act, untrammeled by any particular course of things. It is deflected neither by love, nor by hate, for what in fact comes to pass. The *particularities* of the world presuppose *it*; while *it* merely presupposes the *general* metaphysical character of creative advance, of which it is the primordial exemplification" (pp. 343–44).[4] *This* is the impetus to natural theology. I-thou and I-it conjoined in process. How different from Sherrington was Whitehead's Gifford Lectures!

Nevertheless, two important conclusions regarding Sherrington's lectures remain: first, is that Sherrington's students engaged in wishful thinking when they insisted that their mentor found God when neurology met theology; second, that there is a theistic answer to the atheistic humanism of Sherrington then and Tallis now. It is found in Scruton's proposal of cognitive dualism within a hylomorphic framework of Bracken's trinitarian process theology. The story of natural theology is by no means over.

**Funding:** This research received no external funding.

**Data Availability Statement:** Not applicable.

**Conflicts of Interest:** The author declares no conflict of interest.

## Notes

1.  See Freud's (1940) *Outline of Psycho-Analysis* first published as *Abriss der Psychoanalyse*; (Koffka 1935); (Schrödinger 1935); and C. S. Lewis's (1941) excursion into the demonic that first appeared in serial form with "The Screwtape Letters" in *The Guardian* (2 May 1941–28 November 1941).

2.  Spinoza is usually regarded as a monist in opposition to Descartes, but, in answer to Elmer Ellsworth Powell's (1906) *Spinoza and Religion*, Boodin (see reference) has convincingly argued that his psychic and physical parallelism constitutes an effective dualism.

3.  An occasion is a key term in process theology. When things happen (a ball bouncing, a child crying, or even a star forming) we normally think of them as discrete, fixed events; Whitehead, however, views them as actions or things in and of themselves because they really are experiences of the observer and of the environment of which they are a part; even quanta of energy can be occasions. This is why process thinkers avoid the idea of substances of individual entities that act and are acted upon. For Whitehead, things are not stiff pieces moving—or being moved—about; occasions more accurately reflect the dynamics of nature. Occasions-as-experiences should not be confused with mind or consciousness; to do so would invite panpsychism, a term Whitehead never used. For more, see (Cobb 2015, pp. 13–17, 19).

4.  A complete discussion of natural theology from a process perspective is beyond the scope of this article. A more thorough treatment is available in Cobb's (2007) *A Christian Natural Theology*.

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
