# Peer review of "Neurology Meets Theology: Charles Sherrington’s Gifford Lectures Then and Now"

_religions, doi:10.3390/rel14101310_

Round 1

Reviewer 1 Report

Although the arguments made by the author are cogent and certainly provide a reasoned justification for his/her opinion, I question his/her conclusion that reductionist/non-reductionist descriptors ought to replace the varieties of current labels used to identify the different forms of humanism.  I believe that humanism is far more nuanced than the terms reductionist/non-reductionist suggests. There is the ancient humanism of Socrates and Lucretius; the classical humanism of Petrarch and Erasmus, theistic humanism of Maritan and Niebuhr, naturalistic humanism of Dewey, atheistic humanism of Marx, literary humanism of Thomas More, the evolutionary humanism of Julian Huxley, humanistic mysticism of A. E. Haydon, cosmic humanism of Frank C. Doan and Carl Sagan, etc.  I doubt that these adjectives, which I believe carry significant meaning and value, ought to be replaced by the  simple application of reductionist/non-reductionist terminology. Nevertheless, the article is well written and well reasoned and will no doubt elicit a range of responses from readers of the journal.

What is clear from the article is that the range of opinions identified by the author in the paper are, in effect, a response to the observation Darwin made in Descent that the difference in mind between animals and humans is one of degree rather than kind. It amazes me how, in a universe filled with billions upon billions of stars and planets, that we continue to find ways to describe human evolution as some form of exceptionalism.

Author Response

I agree with reader 1 that humanism is a many-faceted and nuanced worldview. Perhaps I misstated my aims in the original ms., for it was not my intent to suggest that reductionist/non-reductionist categories replace humanism as a general descriptor of one’s attitude and worldview. In fact, the many forms that humanism has taken throughout history suggest that it can mean many things in many contexts. I have no intention of replacing humanism, simply of refining its applications. But since my earlier comments regarding reduction and non-reduction have apparently contributed to confusion, I have walked that position back in this version. Reader 1 is quite correct in pointing out the differences between, for example, the theistic humanism of Reinhold Niebuhr and the atheistic humanism of Karl Marx (or might I add, the late Christopher Hitchens). It is this very diffused nature of humanism that raises questions with regard to natural theology. Thus, it can be said that the reductionist humanism of Albert Einstein presents a very different worldview picture from that of the non-reductionist views of Raymond Tallis and Charles Sherrington. It is interesting to note, however, that Einstein's reductionism had a deleterious effect on his science. Reductive and non-reductive categories remain important aspects of humanism generally and are substantive rather than stylistic, the former probably more problematic than the latter. 

While we can have our differences, it is sincerely encouraging to witness the magnanimity of reader 1. The willingness to let such differences be expressed and debated in a genuinely free marketplace of ideas is something that is sadly becoming less common in this increasingly polarized and balkanized intellectual climate. In short, I greatly appreciate reader 1’s willingness to move this toward publication in the spirit of keeping discussion and debate alive and well.

I will end by agreeing with reader 1 that Darwin’s proposal in Descent of continuity of mind (i.e. the rejection of human exceptionalism) is an important underlying theme of the paper and indeed a significant aspect of the whole mind/body question. The critique by Bolhuis and Wynne in Nature of the persistent overinterpretation of animal behavior is perhaps suggestive of problems underlying Darwin's continuity of mind. In my revised version this is further addressed in my reference to Roger Scruton’s The Face of God. Scruton calls it “human distinctiveness” but he means the same thing. I doubt it will change reader 1’s view, but it is nonetheless further amplified in this revised version.

Since my two main goals remain unchanged: 1) to correct the historiography surrounding Sherrington’s natural theology (a matter of historical fact); and 2) to offer a theistic reply centered on a hylomorphic answer to the mind/body problem (a matter of perspective open to further discussion and debate), I would think that reader 1’s willingness to publish the revised version remains, as no doubt will these honestly, openly, and fairly expressed objections.

Reviewer 2 Report

This is a genuinely interesting and stimulating piece to read. It is well-written and engages with a wide range of important thinkers, past and present, on matters of great significance to the interface between science and religion generally and neurology and natural theology specifically.  And on a figure,  Sherrington, whose work should be more widely known. 

The article suffers from an insufficiently clear sense of identity, however. After introducing the man and his achievements, the article seems to move between responses to Sherrington's Gifford Lectures and the associated publication, analysis of his arguments in those lectures, concerns about how to classify the religious outlook of different thinkers who are or are not close to Sherrington's, while also offering a process-theistic solution to the mind-body problem by way of an 'advance' of Aristotle's and Aquinas's hylomorphism. That's quite a program! The elements overlap, of course, but not in an entirely satisfactory way. 

The abstract notes: 'The essay concludes that religious/atheist labels are ultimately problematic when applied to humanism and naturalism.' If that is the conclusion, then much more consideration should be given earlier in the article to examples where they are used in a way that the author wishes to correct / challenge. Who thinks that 'religious/atheist' labels are ultimately not problematic when applied to the humanism and naturalism of Sherrington et al? The justification for this (I would like to think worthwhile) intervention into scholarship ought to be clearer.

The author should also have been frank earlier on about what constitutes 'religion / 'religious' in their view. The difficulty of specifying this is acknowledged at the end, but that is a bit late in the day. As is the introduction of theological giants like Karl Barth in the final thoughts.

In short, there needs to be a clearer and more integrated sense of purpose to the analsysis leading to a coherent conclusion that reflects the prior discussion more fully. This may mean not trying to achieve quite as much: the hylomorphism proposal seemed to me something that should be kept. It is making a comeback, and the author links it to process thology in an interesting way,

If the author really wants to go down the road of a tick-box chart, assigning religious and philosophical identities to individuals, then I would make that the stated concern at the outset.  I am not sure it adds a great deal.

If the central issue/thesis is that the most important/interesting point of contention in the meaning and interpretation of contemporary science is not whether it allows for religious commitment, but whether it is reductionist or not, then I suggest making that a sustained part of the argument earlier on. 

Finally, the 'fallibilist' epistemology mentioned in the abstract, in connection with 'warranted assessability', ought to have been more of a consideration in the analysis prior to the conclusion. What are the criteria for such a warrant? I wasn't clear on that. 

Some additional suggested referencess

The rejection of Descartes's substance dualism, but sympathy for conceptual dualism - and rejection of crude materialism - seems to suggest some common ground with Spinoza, whose Ethics may be a relevant reference point for this essay. 

The late philosopher Roger Scruton wrote many fine works on the topics the author engages with. I am sure we can do better than an essay in The New Atlantic. You might start with his own Gifford Lectures and book: The Face of God (Continuum 2014). 

There was an excellent online resource on the Gifford lectures, where the interested could read the full will of Lord Gifford and his definition of natural theology. It is currently undergoing reconstruction. But part of the will can be found here: https://gifford.wp.st-andrews.ac.uk/lord-gifford-and-his-will/

An academic journal in Scotland (Theology in Scotland) has started what promises to be a long-running series of articles on The Gifford Lectures. One of the themes in the first two (the first is an overview of the whole enterprise) is the failure of so many speakers to actually speak to the theme of the lectures:

Jonathan C P Birch, 'Imagining the Gifford Lectures: 134 Not Out', Theology in Scotland (29.1), Spring 2022, pp. 55-71: https://ojs.st-andrews.ac.uk/index.php/TIS/article/view/2433 

Jonathan C P Birch, 'The Theological House that Jack (Un)Built: Halberstam on an aesthetics of collapse and mushrooms among the ruins', Theology in Scotland (29.2), Autumn 2022, pp. 55-71: https://ojs.st-andrews.ac.uk/index.php/TIS/article/view/2433 

Please see the attached PDF with more concrete and specific comments, suggestions, and corrections

The article is well written. My occasional observations on the text can be found on the attached PDF.

Author Response

I am glad reader 2 enjoyed this essay, finding it “interesting and stimulating.” Having read and carefully considered the comments, I find myself largely in agreement. In its original form, the paper did indeed seem to suffer from a lack of “clear identity.” In fact, much of this was caused by cluttering the ms. with extraneous issues such as the imposition of a chart delineating belief and the introduction of “warranted assertibility” late in the conclusion. They are removed in this revised ms. I also have pulled back on my claims for reductive and nonreductive humanism. Reductionism is an important part of the discussion, but it is not a central factor in belief (witness the reductive “neuromania” of Churchland and reductive determinism of Einstein’s respective atheisms against the nonreductive atheisms of  Sherrington and Tallis). Rather than retaining these and moving them earlier in the paper as suggested, however, I accept reader 2’s criticism that the essay tries to do too much and have either removed them altogether or pulled back on my claims because they detract from my two main aims: 1) to correct the historiography surrounding Sherrington’s natural theology; and 2) to present a theistic reply based upon the hylomorphic answer to the mind/body problem. I was particularly pleased to see that reader 2 noted this second aim approvingly.

In order to accomplish these goals, I have implemented reader 2’s most helpful suggestions in the following ways:

  • I have inserted some discussion of Spinoza along with an explanatory note.
  • I have expanded my references to Roger Scruton. Reader 2 is right, “we can do better than an essay in the New Atlantic.” But that New Atlantic essay was not a synopsis of The Face of God (2012) but rather his later book, The Soul of the World (2014). Since I fully share reader 2’s admiration of Scruton’s work, I have happily included discussion of both (The Face of God because it represents Scruton’s Gifford Lectures [2010] that frame an effective reply to Sherrington and Tallis generally, and The Soul of the World because it is here that Scruton introduces hylomorphism).
  • I also appreciate reader 2 alerting me to the work of Jonathan C. P. Birch. His discussion of the Gifford Lectures provides interesting and useful historical contexts to the series that fit in well with the Sherrington discussion here.
  • I have gone through the highlighted portions of the text and in most cases made the necessary corrections and changes.

I hope the revised version of the ms. meets with reader 2’s approval. I have followed the suggestions for improvement with appreciation for all the constructive criticism. Reader 2 will find a new abstract that gives clearer focus to the paper as whole, and while the arguments haven’t substantively changed, they are I think more clearly and coherently expressed buttressed by the scholarship hereby recommended for which I am grateful.